# Optically Guided High-Frequency Ultrasound Shows Superior Efficacy for Preoperative Estimation of Breslow Thickness in Comparison with Multispectral Imaging: A Single-Center Prospective Validation Study

**DOI:** 10.3390/cancers16010157

**Published:** 2023-12-28

**Authors:** Noémi Nóra Varga, Mehdi Boostani, Klára Farkas, András Bánvölgyi, Kende Lőrincz, Máté Posta, Ilze Lihacova, Alexey Lihachev, Márta Medvecz, Péter Holló, Gyorgy Paragh, Norbert M. Wikonkál, Szabolcs Bozsányi, Norbert Kiss

**Affiliations:** 1Department of Dermatology, Venereology and Dermatooncology, Semmelweis University, 1085 Budapest, Hungary; varga.noemi@stud.semmelweis.hu (N.N.V.); mehdi_parsii@yahoo.com (M.B.); farkas.klara@phd.semmelweis.hu (K.F.); banvolgyi.andras@med.semmelweis-univ.hu (A.B.); lorincz.kende@med.semmelweis-univ.hu (K.L.); medvecz.marta@med.semmelweis-univ.hu (M.M.); hollo.peter@med.semmelweis-univ.hu (P.H.); wikonkal.norbert@med.semmelweis-univ.hu (N.M.W.); szabolcs.bozsanyi@roswellpark.org (S.B.); 2Systems Biology of Reproduction Research Group, Institute of Molecular Life Sciences, HUN-REN Research Centre for Natural Sciences, 1117 Budapest, Hungary; posta.mate@ttk.hu; 3Biophotonics Laboratory, Institute of Atomic Physics and Spectroscopy, University of Latvia, 1004 Riga, Latvia; ilze.lihacova@lu.lv (I.L.); aleksejs.lihacovs@lu.lv (A.L.); 4Department of Dermatology, Roswell Park Comprehensive Cancer Center, Buffalo, NY 14203, USA; gyorgy.paragh@roswellpark.org

**Keywords:** melanoma, Breslow thickness, diagnosis, multispectral imaging, high-frequency ultrasound, histology

## Abstract

**Simple Summary:**

In this study, we focused on melanoma, a highly aggressive form of skin cancer with a rising global incidence. Melanoma’s staging and treatment depend on Breslow thickness, which is usually unavailable at the initial diagnosis. We aimed to compare two novel imaging techniques, optically guided high-frequency ultrasound (OG-HFUS) and multispectral imaging (MSI), to estimate Breslow thickness and improve preoperative staging. We enrolled 101 patients with confirmed primary melanoma, categorized by tumor thickness, and utilized these imaging methods. Our findings revealed that OG-HFUS was superior compared to MSI in estimating Breslow thickness, making it a valuable tool for melanoma diagnosis and patient care. This research may enhance preoperative staging and treatment decisions in the field of melanoma, offering promising implications for patient outcomes.

**Abstract:**

Melanoma is the most aggressive form of skin cancer that is known for its metastatic potential and has an increasing incidence worldwide. Breslow thickness, which determines the staging and surgical margin of the tumor, is unavailable at initial diagnosis. Novel imaging techniques for assessing Breslow thickness lack comparative data. This study evaluates optically guided high-frequency ultrasound (OG-HFUS) and multispectral imaging (MSI) for preoperative estimation of Breslow thickness and staging. We enrolled 101 patients with histologically confirmed primary melanoma and categorized them based on tumor thickness. Optically guided 33 MHz HFUS and MSI were utilized for the assessment. Our MSI-based algorithm categorized melanomas into three subgroups with a sensitivity of 62.6%, specificity of 81.3%, and fair agreement (κ = 0.440, CI: 0.298–0.583). In contrast, OG-HFUS demonstrated a sensitivity of 91.8%, specificity of 96.0%, and almost perfect agreement (κ = 0.858, CI: 0.763–0.952). OG-HFUS performed better than MSI in estimating Breslow thickness, emphasizing its potential as a valuable tool for melanoma diagnosis and patient management. OG-HFUS holds promise for enhancing preoperative staging and treatment decision-making in melanoma.

## 1. Introduction

Malignant melanoma is a highly aggressive form of skin cancer originating from melanocytes. It has witnessed a global surge in its incidence, emphasizing the necessity for vigilant skin examinations and sun protection [1]. The epidemiology of malignant melanoma reveals varying rates across populations and regions, notably higher in areas with increased exposure to ultraviolet (UV) radiation from the sun [1]. Factors such as fair skin, a history of sunburns, familial predisposition, and excessive UV exposure significantly contribute to the development of melanoma [2,3]. Understanding its diverse subtypes and stages is crucial for accurate diagnosis and tailored treatment strategies [4,5]. Timely intervention is paramount, emphasizing the significance of proactive skin health measures and awareness to combat this potentially life-threatening condition [6,7]. Efforts focusing on sun protection advocacy, routine skin screenings, and early detection play a pivotal role in its epidemiology, stressing prevention as a critical measure in reducing its incidence [8,9].

Accurate pathological staging of melanoma relies on histological results and serves as a crucial component of the diagnostic process. The histological examination allows for a detailed evaluation of the tumor’s characteristics, including its architectural features, cell type, mitotic rate, and, importantly, the Breslow depth. These factors play a significant role in determining the accurate stage of melanoma and enabling healthcare professionals to make informed decisions regarding the most appropriate treatment approach [10,11,12]. Although non-invasive imaging techniques can offer valuable insights, they are limited in determining the histological parameters necessary for accurate staging [10,11].

Several imaging technologies have been investigated as potential non-invasive diagnostic tools for assessing Breslow thickness, such as dermoscopy (DSC) [13], multispectral imaging (MSI) [14], high-frequency ultrasound (HFUS) [15], and optical coherence tomography (OCT) [16].

Dermoscopy is a non-invasive imaging technique that magnifies skin lesions, revealing intricate details not visible to the naked eye. By utilizing specialized lights and filters, dermoscopy allows dermatologists to observe pigment patterns, structures, and vascular features crucial for diagnosing skin conditions like melanoma [17,18]. Dermoscopy has become an invaluable non-invasive tool in dermatology, particularly in the early detection of melanoma. It has revolutionized the diagnosis of melanoma by meticulously examining skin lesions and unveiling unique features that are pivotal in diagnosing melanoma at its nascent stages [19]. Specific patterns and structures emerge through dermoscopy, aiding in the differentiation between benign and malignant lesions, such as irregular pigment network, asymmetrical structures, abrupt peripheral streaks, and various shades of color variation [19]. Dermoscopy also reveals subtle nuances that often evade the naked eye but are distinctive under dermoscopic analysis, such as the presence of atypical vessels, regression structures, and the uneven distribution of colors within a lesion, which are significant indicators guiding accurate diagnoses and prompt interventions in suspected cases of melanoma [19].

Multispectral imaging (MSI) is an emerging imaging technique in dermatology that collects and analyzes reflected radiation from different wavelength bands in the visible (400–700) and even beyond the visible light range, using light bulbs or LED lights [14,20,21,22,23]. The MSI name is used for techniques employing ten or less than ten spectral bands; modalities using more than ten are called hyperspectral imaging (HSI) techniques [23]. The main advantage of this technique is the combination of spectrophotometry (spectral resolution) and digital cameras (spatial resolution) [14,24], visualizing the light–tissue interactions of the skin [23]. The skin has many chromophores which interact with the irradiating light, including melanin, oxy- and deoxyhemoglobin, bilirubin, and collagen [25], allowing the user to study these components in different physiologic and pathologic conditions. It is also notable that MSI is cost-effective and can be used with smartphone cameras [14]. Our research group developed an algorithm previously using MSI to estimate the Breslow tumor depth of melanoma using three different wavelength bands (green, red, and infrared) and a multistep sorting algorithm into three distinct subgroups: 1. Breslow thickness under 1 mm, 2. Breslow thickness between 1 and 2 mm, 3. Breslow thickness over 2 mm. The algorithm could sort melanomas into these three subgroups with a sensitivity of 78.00% and specificity of 89.00% and a substantial agreement (κ = 0.67; 95% CI, 0.58–0.76) [26].

In dermatology, HFUS has gained attention as a non-invasive diagnostic tool for skin cancer. Our lab has studied the use of HFUS in differentiating high-risk from low-risk basal cell carcinomas [27]. This imaging technique, also known as high-resolution ultrasound, employs sound waves with frequencies greater than 20 megahertz (MHz) to generate images of tissue structures [15]. Melanoma is generally characterized by hypoechogenic, homogenous, and well-defined lesions on HFUS images [28]. Using HFUS, the depth of the melanoma can be measured by determining the distance between the top layer of the skin and the deepest point of the tumor. Therefore, HFUS may be capable of preoperative diagnosis of melanoma [29] and estimation of Breslow thickness [28].

OCT operates on the principle of low-coherence interferometry, emitting light waves that reflect off skin tissue. By measuring the echo time delay and intensity of the reflected light, it creates detailed images based on the varying optical properties within the skin’s layers [30,31]. In the context of melanoma detection, OCT offers remarkable significance by providing detailed insights into the skin’s architecture, allowing for the visualization of structural changes within lesions. This technology significantly aids in distinguishing between benign and malignant lesions, enabling clinicians to make more precise and timely diagnoses in suspected cases of melanoma [32,33]. Furthermore, OCT imaging offers a non-invasive means to diagnose and monitor the development of various skin conditions over time, serving as a viable substitute for skin biopsies or surgical interventions for different skin conditions [34,35,36].

Considering the increasing incidence of melanoma and the need for faster diagnostic, staging, and therapeutic procedures, this study aims to compare the efficacy of novel optically guided high-frequency ultrasound (OG-HFUS) and MSI in preoperative estimation of Breslow tumor thickness.

## 2. Materials and Methods

### 2.1. Inclusion and Exclusion Criteria

In this Single-Center Prospective Validation Study, the inclusion criteria involved obtaining informed consent from patients and validating primary cutaneous melanomas through histopathological confirmation by expert dermatopathologists. Eligibility was restricted to melanomas exhibiting a Breslow thickness of less than 10 mm.

Exclusion criteria were applied to cases with in situ melanomas or melanoma metastases, and primary melanomas located in specific anatomical sites (such as acral, genitalia, or mucosa). Furthermore, lesions presenting extensive hair, bleeding, or scaling that could impede accurate imaging were also excluded.

### 2.2. OG-HFUS Device and Image Analysis

We conducted measurements at the Department of Dermatology, Venereology, and Dermatooncology, Semmelweis University, using the portable OG-HFUS device (Dermus SkinScanner, Dermus Ltd., Budapest, Hungary). This device combines ultrasound imaging and optical image capture capabilities. The OG-HFUS device has a single-element ultrasound transducer operating at a nominal center frequency of 33 MHz ranging between 20–40 Mhz. A silicon membrane covers the imaging window, and for scanning, a gel is applied by the researcher, who uses an optical imaging module for positioning the lesion. The resulting ultrasound image is presented in a color scale format to enhance contrast. The optical image obtained by the device offers a field of view measuring 15 mm × 15 mm. The ultrasound image extends laterally up to 12 mm and reaches a maximum penetration depth of 10 mm. In cases where the lesion size exceeded the optical field of view, we captured additional image sets to ensure comprehensive coverage and analysis. Both the optical and ultrasound images are then stored using a cloud-based system, which saves all the photos under a patient ID with the birth date, sex, diagnosis, and exact location of the body region [37]. These captured ultrasound images were used to measure the presumed tumor thickness. The depth measurement of tumor thickness was obtained within 1 min after capturing the ultrasound image.

### 2.3. MSI Device and Image Analysis

The measurements were conducted at the Department of Dermatology, Venereology, and Dermatooncology at Semmelweis University, Budapest, Hungary. The handheld prototype utilized in this study was jointly developed by the University of Latvia and Riga Technical University (Riga, Latvia). The illumination source consisted of a LED ring comprising four types of LED diodes emitting at wavelengths of 405 nm (for Autofluorescence/AF excitation), 525 nm (Green/G), 660 nm (Red/R), and 940 nm (Infrared/IR), enabling penetration into different skin layers. The LEDs provided an irradiating power density of 20 mW/cm^2^ and had a field of view of 2 × 2 cm^2^. For quantitative analysis, we utilized the G, R, and IR channels. The device was designed to capture skin diffuse reflectance images using the fixed arrangement of the four LEDs at 35 mm. These LEDs were arranged circularly within the ring and covered by a matte plate diffuser to ensure uniform illumination. Image acquisition was performed using a 5-megapixel color CMOS IDS camera (MT9P006STC, IDS uEye UI3581LE-C-HQ, Obersulm, Germany) placed at a distance of 60 mm from the illuminated skin [38]. The acquired images were automatically transferred to a cloud server for further data processing and analysis [39]. To capture the AF emission images and block the 405 nm excitation illumination, a long-pass filter (T515 nm > 90%) was placed in front of the camera. In cases where the lesion size exceeded the camera’s field of view, we captured additional image sets to ensure comprehensive coverage and analysis. A detailed description of this prototype device was previously published [38,40]. The LED-based multispectral images were analyzed using ImageJ v1.46 software (NIH, Bethesda, MD, USA) [41]. For intensity analysis and shape description, we manually selected regions of interest (ROIs) using Freehand selections. ROIs corresponding to melanomas were recorded using the ROI manager function of the ImageJ software, ensuring that the analyzed area was consistent across all channels (G, R, and IR). Mean gray value (integrated density/area), circularity (4π × area/perimeter^2^), solidity (area/convex area), and roundness (4 × area/(π × major_axis^2^)) were measured as described in our previous publication [26]. The entire imaging procedure and subsequent thickness estimation are completed within a few minutes.

### 2.4. Melanoma Classification Algorithm

In our previous research [26], we developed an algorithm to classify melanomas into three clinically relevant subgroups (Breslow thickness < 1 mm, Breslow thickness between 1–2 mm, and Breslow thickness > 2 mm) based on the shape descriptors and intensity values of their MSI images. This modified melanoma classification algorithm (Figure 1.) was used in the current study to assess the efficacy of MSI. The first step of the original algorithm was not used in this research.

### 2.5. Statistical Analysis

We compared our measurements on the captured ultrasound images with histologically determined Breslow thickness to assess the accuracy of OG-HFUS. To describe the correlation, we employed Pearson correlation and calculated the reliability within each clinical category when classifying patients based on ultrasound results.

We developed a multivariate linear model to analyze Pearson correlation between multispectral imaging (utilizing green, red, and infrared channels) and Breslow thickness in skin lesions. Circularity data were also considered for a comprehensive evaluation of morphological aspects. Significance evaluation was conducted using F statistics.

For statistical evaluation, we used the scikit-learn, scipy, and statmodels libraries in the Python programming language and environment. We determined the sensitivity, specificity, positive predictive value (PPV), negative predictive value (NPV), and mean squared error (MSE) in each clinical category. For overall model estimation, we presented the micro-averaged sensitivity, specificity, positive predictive, and negative predictive values. Cohen’s kappa (κ) was used to calculate concordance. *p* values below 0.05 were considered statistically significant. The results are expressed as mean +/− standard deviation.

## 3. Results

### 3.1. Patient Data

In total, a cohort of 101 melanoma patients was investigated, characterized by a mean age of 64.20 ± 15.24 years, comprising 53 males and 48 females, resulting in a sex ratio of 52% males and 48% females These evaluated primary cutaneous melanomas were distributed across various anatomical regions: 60 on the trunk, 32 on the extremities, 6 on the cheek, 2 on the forehead, and 1 on the neck. The analysis revealed that the mean Breslow thickness was 1.61 mm ± 1.69 mm. The range of lesion depth was quite substantial, ranging from a minimum thickness of 0.135 mm to a maximum of 8.12 mm. The detailed breakdown of melanoma subtypes is presented in Table 1.

### 3.2. Diagnostic Accuracy of the OG-HFUS

The overall performance of the OG-HFUS demonstrated substantial reliability, as indicated by Cohen’s kappa coefficient of 0.858 (95% CI: 0.763–0.952). The HFUS imaging results can be visualized in Figure 2. This value suggests almost perfect agreement between non-invasive ultrasound measurements and the gold standard histological data. Measurements with the OG-HFUS exhibited a significant positive correlation with histological Breslow thickness (r = 0.943, *p* < 0.0001) (Figure 3). The detailed performance metrics of OG-HFUS for estimating Breslow thickness are presented in Table 2.

### 3.3. Diagnostic Accuracy of the MSI

The overall performance of the MSI demonstrated fair agreement, as Cohen’s kappa coefficient became 0.440 (95% CI: 0.298–0.583). The MSI imaging results can be visualized in Figure 4. Independently, among the shape descriptors and intensity values of different channels, the IR channel exhibited the highest correlation (r: −0.659, *p* < 0.0001) with Breslow thickness. However, multivariate correlation analysis revealed a strong correlation between the multivariate data and Breslow depth (r: 0.714, *p* < 0.0001) (Figure 5). It is important to note, however, that this predictive value lags behind the performance of OG-HFUS. The detailed efficacy of MSI estimating Breslow thickness can be seen in Table 3.

### 3.4. Comparative Analysis of Imaging Methods

In our comparative analysis, OG-HFUS demonstrated superior performance in the preoperative estimation of Breslow thickness. OG-HFUS achieved a sensitivity of 91.8%, a specificity of 96.0%, and exhibited almost perfect agreement (κ = 0.858, CI: 0.763–0.952). In contrast, MSI showed a sensitivity of 62.6%, and specificity of 81.3%, with a fair agreement (κ = 0.440, CI: 0.298–0.583). OG-HFUS also displayed a positive predictive value (PPV) of 91.8% and a negative predictive value (NPV) of 96.0%, surpassing the PPV and NPV of MSI, 62.6% and 81.3%, respectively. These findings underscore the improved accuracy and efficacy of the OG-HFUS approach in determining Breslow thickness compared to MSI.

## 4. Discussion

Breslow tumor thickness is a critical predictor of survival among patients with localized melanoma, [42] and this measurement defines the optimal safety margin for treatment [5]. If the surgical margin is insufficient, reoperation becomes necessary. Conversely, when it exceeds what is required, it may result in a larger scar and potential loss of function in the affected area, imposing an unwanted burden on patients [43,44]. Treating melanomas in the head and neck region can be even more challenging, and achieving excision with an appropriate surgical border is not always feasible [45]. Our findings revealed that OG-HFUS was superior compared to MSI, demonstrating higher sensitivity, specificity, and overall agreement in predicting Breslow thickness.

Preoperative HFUS is a promising tool for aiding surgeons in single-step melanoma surgery [46,47]. Particularly, HFUS, in combination with digital dermoscopy, enhances accuracy, and videomicroscopy can differentiate between thick and thin melanomas with a sensitivity of 86.7% [48]. Optical coherence tomography has also shown promise in predicting melanoma tumor thickness based on vascular morphology [49]. Reflectance confocal microscopy (RCM) is an emerging dermatological tool with cellular-level resolution and a penetration depth of approximately 200 μm [50,51]. It has proven to be an accurate modality for presurgical margin mapping of LMMs [52,53] but it is not suitable for estimating Breslow tumor thickness, especially in deep lesions. Moreover, Verzì et al.’s study illustrates the potential of LC-OCT to enhance the non-invasive diagnosis of LMM, an in situ melanoma subtype prevalent in elderly individuals. Investigating the LC-OCT features of an LMM lesion on a 49-year-old’s nose, the technique revealed characteristic microscopic traits in both horizontal and vertical imaging. LC-OCT clearly depicted large, bright roundish, or dendritic atypical cells around hair follicles and within the epidermis, signaling atypical melanocytes and their tendency towards folliculotropism. Despite being based on a single case, the strong correlation between LC-OCT images and histopathological sections underscores the potential of LC-OCT in facilitating non-invasive LMM diagnoses [54]. Interestingly, there is a growing trend in combining various imaging modalities, such as line-field optical coherence tomography (LC-OCT), which combines the cellular-level resolution of RCM with the depth penetration of OCT [55]. These combined modalities are well-suited for analyzing melanomas and their structural characteristics. Barragán-Estudillo et al. established links between specific dermoscopic and RCM findings with histopathologic parameters in primary melanomas. Notably, the presence of pigmented networks in dermoscopy was linked to lower Breslow and mitotic rates (*p* = 0.002), while visible vessels correlated with higher Breslow and mitotic indexes (*p* = 0.001). Furthermore, confocal identification of dermal nests or atypical cells in the papillary dermis related to elevated mitotic rates (*p* = 0.006 and *p* = 0.03). Moreover, superficial dermal inflammation visible through confocal imaging was associated with higher Breslow and mitotic index (*p* = 0.04). These findings indicate the potential of combined dermoscopy and RCM assessments in estimating primary melanoma characteristics and predicting crucial histopathologic parameters [56]. Furthermore, Fedorov et al. have used an OCT-HFUS-Raman spectroscopy combined tool to virtually biopsy melanomas, harnessing the advantages of these different modalities [57]. Moreover, Suppa et al.’s comprehensive review of LC-OCT’s application in melanocytic lesions showcases its groundbreaking use in imaging melanoma, highlighting details like epidermal invasion and disruptions in the dermal-epidermal junction. The review defines criteria for benign lesions, compares LC-OCT with reflectance confocal microscopy, and finds similar diagnostic performance in identifying melanoma. It emphasizes LC-OCT’s correlation with histopathology/RCM in recognizing critical features in diverse lesions, including in situ and invasive melanomas, Spitz nevi, and genital pigmented lesions. Case reports underscore LC-OCT’s ability to detect cellular structures akin to histopathology, validating its accuracy. Lastly, the review anticipates advanced LC-OCT devices featuring global dermoscopic colocalization, underlining its pivotal role in precise lesion characterization, aligning seamlessly with established diagnostic criteria [58]. The discussion on the reliability of non-invasive techniques, such as LC-OCT and RCM, concerning their strong correlation with both vertical and horizontal histopathology in clinical practice is crucial. The existing literature, notably the study by Lacarrubba et al., delineates the utility of RCM in identifying key microscopic features of discoid lupus erythematosus (DLE) and its correlation with horizontal histopathological sections. Lacarrubba et al.’s investigation demonstrated RCM’s ability to accurately detect typical inflammatory changes and dermo-epidermal junction alterations characteristic of DLE. This alignment between RCM and histopathology, both in horizontal and vertical views, substantiates the precision of non-invasive imaging techniques and their potential for enhancing clinical diagnoses in conditions like psoriasis, discoid lupus, and other dermatological disorders [51].

Breslow thickness is a critical parameter for melanoma staging and surgical planning. OG-HFUS demonstrated an exceptional sensitivity of 91.8% in estimating Breslow thickness, signifying its ability to accurately identify both thin and thick melanomas. This high sensitivity is invaluable in clinical practice as it reduces the risk of underestimating tumor thickness, which can lead to inadequate surgical margins and potential disease recurrence. In contrast, MSI exhibited a lower sensitivity of 62.6%, implying a higher likelihood of false negatives in Breslow thickness estimation. False negatives can be particularly problematic in melanoma diagnosis as they may result in undertreatment.

OG-HFUS also excelled in terms of specificity, boasting a specificity of 96.0%, indicating a low rate of false positives. The high specificity of OG-HFUS suggests that it can reliably differentiate non-melanoma lesions from melanomas, reducing unnecessary excisions and alleviating patient anxiety. On the other hand, MSI showed lower specificity (81.3%) and a fair agreement (κ = 0.440). The decreased specificity of MSI could result in more false-positive findings, potentially leading to unnecessary excisions and increased healthcare costs. This specificity is lower compared to our previous findings, where the MSI algorithm could sort melanomas into three subgroups with a sensitivity of 78.00% and specificity of 89.00% [26]. The difference in specificity might be attributed to variances in the study cohorts. It is also important to note that future improvements of this algorithm could further improve the performance of this evaluation. 

When comparing the methods and results of our study with those of Kaikaris et al., notable differences and outcomes are observed. Kaikaris’ study, conducted between January 2004 and October 2008, utilized a linear 14 MHz frequency ultrasound sensor to measure melanoma depth in 100 patients diagnosed with stage I–II cutaneous melanoma. Their findings demonstrated varying mean differences in measurements, with a higher mean difference of 60 μm observed in tumors matching pathological stage 1 and pathological stage 2 categories. Notably, a stronger correlation (Pearson’s correlation coefficient, r: 0.869; *p*-value < 0.0001) was found in thicker melanomas exceeding 2 mm (mm), while a lower correlation (r: 0.283) was observed in thinner melanomas (1–2 mm) [59].

Comparing our study with Botar et al.’s research reveals differing approaches and findings in the assessment of cutaneous melanoma. Botar’s study, conducted between September 2011 and January 2015, focused on 42 melanoma lesions, using 40 MHz sonography and strain elastography to assess gray-scale features and elastographic characteristics. Their results highlighted the consistent appearance of melanoma lesions, indicating a mean difference between Breslow index and ultrasound thickness of −0.05 mm, suggesting a minimal variance between these measurements. Notably, a strong correlation was identified between elastographic appearance and strain ratios for various tissue interfaces, with a *p*-value indicating statistical significance (*p* < 0.002) [60].

In Reginelli’s investigation, we encounter variations in both the methodologies employed and the resulting findings, particularly in the context of evaluating melanocytic lesions, particularly those exhibiting features indicative of nodular melanoma. Reginelli and colleagues analyzed a total of 14 lesions characterized by distinct dermatoscopic features suggestive of nodular melanoma. Among these, seven lesions underwent excisional biopsy as part of the study. Their methodology centered on the use of ultrasounds equipped with HFUS probes spanning a range from 50 MHz to 70 MHz. The primary aim was to identify the presence of the two hyperechogenic laminae separated by a hypo/anechoic space, a feature used to assess lesion thickness. Notably, Reginelli’s study revealed a consistent positive agreement between the thickness determined through ultrasound and that derived from biopsy in all cases. It is worth highlighting that this analysis encompassed a comparison of seven lesions to calculate thickness, including some designated as satellite lesions (situated less than two centimeter distance from the primary lesion) and in transit lesions (localizable to more than two centimeters from the primary lesion) [61].

In a comparative analysis of our study with Oranges et al., differences in methodology and findings emerge with regards to the evaluation of melanomas through ultrasound techniques. Oranges’ study focused on 27 melanomas in a population with a mean age of 57.6 years, predominantly employing HFUS with a 70 MHz probe to examine lesions before surgical removal. Their approach involved thorough assessments by two skilled and blinded operators using specialized software for repetitive measurements. The findings showed these melanomas to appear as band-like or oval/fusiform-shaped hypoechoic inhomogeneous lesions, with evident intralesional hair follicles in some cases. Comparing ultrasonographic thickness to Breslow thickness revealed variances across cases, with thickness values being either less than, greater than, or identical to Breslow thickness. Notably, Oranges et al.’s study exhibited a low bias and high correlation between Breslow index values and ultrasound measurements for both operators, underscoring the reliability of their approach [62].

Although the duration of the imaging procedure from initial imaging to final thickness estimation slightly differs between HFUS and MSI, taking about 1 min for HFUS and approximately 3 min for MSI, the superior performance of OG-HFUS in estimating Breslow thickness holds important clinical implications. Accurate Breslow thickness assessment is vital for melanoma staging and the determination of surgical margins. OG-HFUS has the potential to enhance preoperative diagnosis, reducing the need for re-excision due to underestimation of tumor thickness. This not only improves patient outcomes but also contributes to cost-effective healthcare delivery by minimizing unnecessary procedures. It is important to acknowledge the limitations of this study. The sample size, while sufficient for preliminary assessment, could be expanded in future research to further validate the findings. Additionally, the study focused on primary melanomas, and the performance of OG-HFUS and MSI in assessing metastatic lesions remains to be explored. Further research could also investigate the cost-effectiveness of these imaging modalities in clinical practice.

## 5. Conclusions

In conclusion, our research evaluated the performance of OG-HFUS and MSI in estimating Breslow thickness and determining the preoperative staging of malignant melanoma. The results demonstrated that OG-HFUS outperformed MSI in diagnostic accuracy with a strong positive correlation with Breslow thickness. This highlights the potential of OG-HFUS as a valuable tool for melanoma diagnosis and patient management. In addition, the preoperative staging with OG-HFUS may be crucial due to the impact of primary excision on SLNB examination. Additionally, MSI analysis of tumor thickness provided the ability to classify melanomas into clinically relevant subgroups. Both imaging methods hold promise for enhancing preoperative staging and treatment decision-making, offering valuable insights for improving outcomes in melanoma patients. However, further research and clinical validation are warranted to establish their broader utility and integration into routine clinical practice.

## Figures and Tables

**Figure 1 cancers-16-00157-f001:**
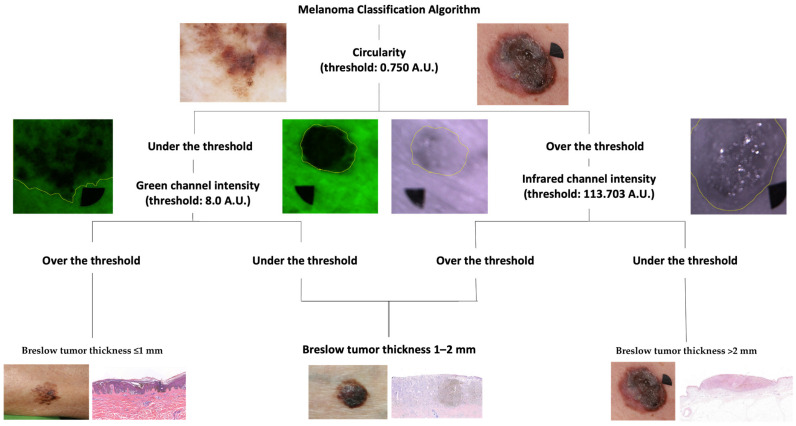
The melanoma classification algorithm. Our algorithm is based on the shape descriptors and the intensity values of the lesion to classify them algorithmically into different Breslow thickness subgroups. The Circularity is the first parameter that sorts the lesions into two subcategories, over and under the threshold (0.75 A.U.). It is followed by the next steps on different branches in the algorithm, where the lesions under the threshold are differentiated into two subgroups based on their intensity in the Green (G) channel, sorting them into two categories, under and over the threshold (8.0 A.U.). In contrast, the previously distinguished subgroup over the Circularity threshold goes through a similar process using the Infrared (I.R.) intensity, sorting them into two subgroups with a threshold value (113.7 A.U.). The lesions under the threshold of Circularity and over the threshold of G channel intensity are classified as melanomas with Breslow < 1 mm, whereas melanomas with similar Circularity values but lower G intensity are classified as tumors with Breslow between 1–2 mm. The melanomas with higher circularity values (over the 0.75 threshold) are subclassified with the I.R. channel values, and melanomas with lower intensities (threshold 113.7 A.U.) were classified as Breslow > 2 mm and melanomas with higher I.R. intensities were classified as Breslow between 1–2 mm. A previous version of this melanoma classification study was published in our earlier study [26], which has been updated. A.U.: Arbitrary Unit. Yellow lines display how the lesions were manually outlined for further analysis.

**Figure 2 cancers-16-00157-f002:**
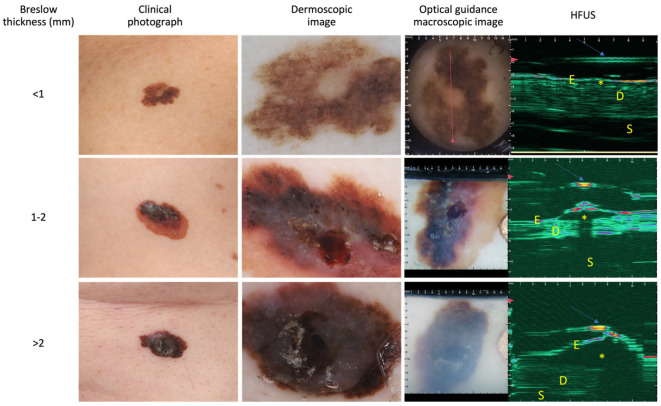
Ultrasound images of distinct melanoma lesions stratified by optically guided high-frequency ultrasound into three groups based on Breslow thickness: <1 mm (row 1), 1–2 mm (row 2), and >2 mm (row 3). Asterisks (*) represent the tumor and arrows indicate the membrane. E: epidermis, D: dermis, S: subcutis. HFUS: high-frequency ultrasound.

**Figure 3 cancers-16-00157-f003:**
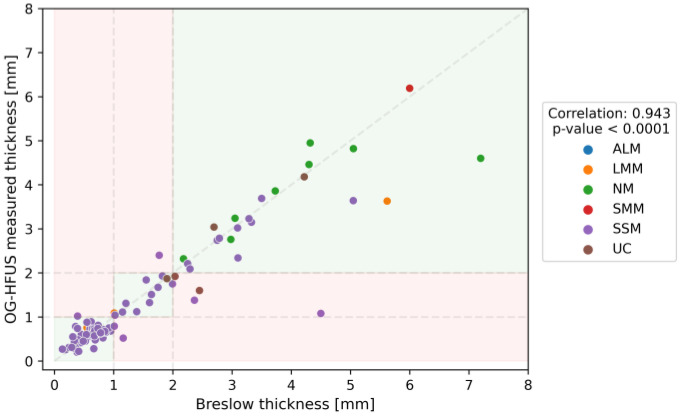
Pearson Correlation Analysis of OG-HFUS Measured Thickness (mm) on the Y-Axis and Histology-Confirmed Breslow Thickness (mm) on the X-Axis, Demonstrating a Significant Positive Correlation with Histological Breslow Thickness (r = 0.943, *p* < 0.0001). Each data point on the graph, represented by a specific color, corresponds to a particular melanoma subtype in the following order: ALM (Acral Lentiginous Melanoma), LMM (Lentigo Maligna Melanoma), NM (Nodular Melanoma), SMM (Superficial Spreading Melanoma), SSM (Superficial Spreading Melanoma) and UC (Unclassified Melanoma). Green background highlights cases where the predicted tumor thickness aligned with the same category (<1mm, 1–2mm, or >2mm) as the actual Breslow thickness determined by pathologists. Conversely, a red background signifies instances where the predicted tumor thickness fell into a different category.

**Figure 4 cancers-16-00157-f004:**
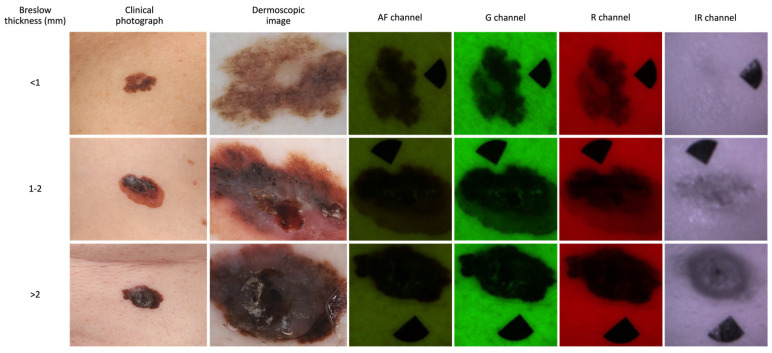
Multispectral images of distinct melanoma lesions classified according to various wavelength channels. Representational images of three lesions segmented into three groups based on Breslow thickness: <1 mm (row 1), 1–2 mm (row 2), and >2 mm (row 3). AF: Autofluorescence, G: Green, R: Red, IR: Infrared.

**Figure 5 cancers-16-00157-f005:**
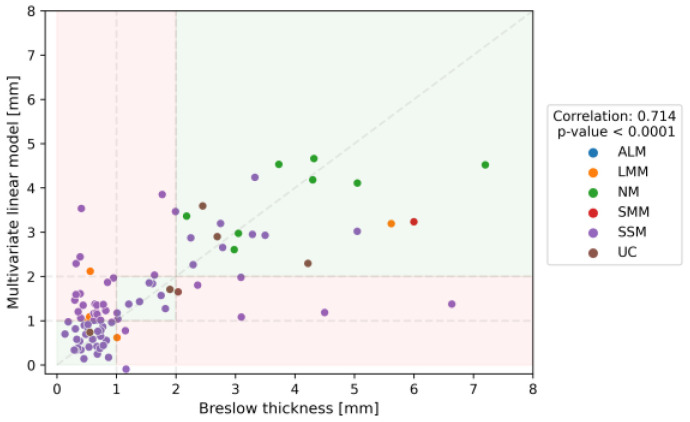
Pearson correlation analysis of multivariate linear model (mm) on the Y-axis and histology-confirmed Breslow thickness (mm) on the X-axis, demonstrating a significant positive correlation with histological Breslow thickness (r = 0.714, *p* < 0.0001). Each data point on the graph, represented by a specific color, corresponds to a particular melanoma subtype in the following order: ALM (Acral Lentiginous Melanoma), LMM (Lentigo Maligna Melanoma), NM (Nodular Melanoma), SMM (Superficial Spreading Melanoma), SSM (Superficial Spreading Melanoma) and UC (Unclassified Melanoma). Green background highlights cases where the predicted tumor thickness aligned with the same category (<1mm, 1-2mm, or >2mm) as the actual Breslow thickness determined by pathologists. Conversely, a red background signifies instances where the predicted tumor thickness fell into a different category.

**Table 1 cancers-16-00157-t001:** Melanoma subtype distribution. The table displays the number of cases for each melanoma subtype, along with their respective percentages in the study.

Subtype	Lesion (*n*)	Distribution Ratio (%)
SSM	69	68.32
NM	8	7.92
SSM sec. Nod.	10	9.90
LMM sec. Nod.	1	0.99
LMM	6	5.94
ALM	1	0.99
UC	4	3.96
Naevoid	2	1.98

SSM: Superficial Spreading Melanoma; NM: Nodular Melanoma; SSM sec. Nod.: Superficial Spreading Melanoma with secondary Nodular component; LMM sec. Nod.: Lentigo Maligna Melanoma with secondary Nodular component; LMM: Lentigo Maligna Melanoma; ALM: Acral Lentiginous Melanoma; UC: Unclassified Melanoma.

**Table 2 cancers-16-00157-t002:** Diagnostic accuracy of OG-HFUS for estimating Breslow thickness.

Breslow (mm)	Patients (*n*)	MSE	Sensitivity (%)	Specificity (%)	PPV (%)	NPV (%)	Cohen’s Kappa (κ)	95% CI
<1	56	0.034	98.2	95.2	96.5	97.6	0.937	0.867–1.000
1–2	15	0.080	80.0	94.0	70.6	96.3	0.701	0.503–0.900
>2	27	1.02	85.2	98.6	95.8	94.6	0.868	0.755–0.980
Total	98	0.31	91.8	96.0	91.8	96.0	0.858	0.763–0.952

MSE: Mean Squared Error, PPV: Positive Predictive Value; NPV: Negative Predictive Value; CI: Confidence Interval.

**Table 3 cancers-16-00157-t003:** Diagnostic accuracy of MSI for estimating Breslow thickness.

Breslow (mm)	Patients (*n*)	MSE	Sensitivity (%)	Specificity (%)	PPV (%)	NPV (%)	Cohen’s Kappa (κ)	95% CI
<1	56	0.64	55.4	93.0	91.2	61.5	0.457	0.286–0.627
1–2	15	0.61	60.0	67.9	25.0	90.5	0.177	−0.052–0.406
>2	28	3.36	78.6	90.1	75.9	91.4	0.680	0.517–0.842
**Total**	99	1.41	62.6	81.3	62.6	81.3	0.440	0.298–0.583

MSE: Mean Squared Error; PPV: Positive Predictive Value; NPV: Negative Predictive Value; CI: Confidence Interval.

## Data Availability

The data that support the findings of this study are available from the corresponding author, N.K., upon reasonable request.

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
