# Peer review of "Optically Guided High-Frequency Ultrasound Shows Superior Efficacy for Preoperative Estimation of Breslow Thickness in Comparison with Multispectral Imaging: A Single-Center Prospective Validation Study"

_cancers, 2023, doi:10.3390/cancers16010157_

Round 1

Reviewer 1 Report

Comments and Suggestions for Authors

The authors submitted a manuscript comparing the two non-invasive imaging devices HFUS and MSI in preoperative thickness measurement of melanomas. Given the high importance of rapid diagnostic assessment, especially in pigmented lesions like melanoma the authors report on a very relevant topic. The work appears to be well-structured and provides clear information. The are only two points I would add:

1) how long does the imaging procedure take from imaging to final thickness estimation (HFUS vs. MSI). Please add few specific details.

2) please add in line 325 the newest work by Suppa M et al, “Line-field confocal optical coherence tomography in melanocytic and non-melanocytic skin tumors”.

Author Response

We extend our gratitude to the reviewer for the invaluable feedback. In response, we have enacted the following refinements:

1-We have provided a description of the imaging procedure durations in the methods section of our manuscript. Additionally, the comparison between HFUS and MSI regarding this aspect has been elaborated upon in the Discussion section as follows “While the duration of the imaging procedure…“ and in the methods (line 160 and 189)

2-We have incorporated the information from the latest work by Suppa M et al., entitled "Line-field confocal optical coherence tomography in melanocytic and non-melanocytic skin tumors" within Our discussion. Specifically, their findings have been seamlessly integrated into our analysis of the characteristics and evaluation of melanoma features as follows “Moreover, Suppa et al.'s comprehensive review of LC-OCT's application in melanocytic lesions…” (line 356)

Incorporating these revisions significantly elevated the manuscript's accuracy and depth, and we're thankful for the valuable guidance offered by the reviewer.

Reviewer 2 Report

Comments and Suggestions for Authors

Noemi Nora Varga, et al. presented a study comparing OG-HFUS to MSI with regard to their accuracy detecting Breslow thickness in pre-surgery mel pts. There are issues that need to be addressed. 

1. The authors should state clearly what this study actually was, e.g., stage II clinical trial, or a prospective cohort study, etc. 

2. It does not make much sense comparing 2 novel techniques, with neither of them as the standard-of-care. It would be more scientifically robust to design this in a more practice-changing way. For example, to demonstrate the decreased rate of re-surgery after applying either method. 

Author Response

We extend our sincere appreciation to the reviewer for the invaluable insights provided. In response, we have implemented the following refinements:

1-We have ensured that the study design is explicitly stated both in the title and and the methods section as follows “In this Single-Center Prospective Validation Study…” (line 126). Two statisticians, one among the authors, Máté Posta, and an independent, Balazs Juhász reviewed the study design and concluded that the term "prospective validation study" should be applied.

2-We thank the reviewer for their insightful suggestion. We agree that a comparative study assessing the reduced rate of re-surgery post-application of either method would be a valuable addition to the field. However, our current study aimed to establish the efficacy between two novel techniques before undertaking a larger practice-changing trial to see which techniques merits further investigation. It is worth noting that we previously conducted a study solely focusing on MSI, and this comparative investigation was intended as a progression toward identifying the more effective technique. We appreciate your feedback and will consider this direction for future research.

The implemented changes notably enriched the manuscript's integrity and thoroughness, and we're grateful for the insightful suggestions shared by the reviewer.

Reviewer 3 Report

Comments and Suggestions for Authors

The authors evaluated two different non-invasive preoperative tools to evaluate the Breslow thickness and staging of cutaneous melanoma, including optically guided high-frequency ultrasound (OG-HFUS) and multispectral imaging (MSI). Their cohort included 101 patients with histologically confirmed primary cutaneous melanoma. Optically guided 33 MHz HFUS and MSI were used for the evaluation of Breslow thickness. Their MSI based algorithm subdivided melanomas into three groups with a sensitivity of 62.6%, specificity of 81.3%, and fair agreement (κ=0.440, CI: 0.298-0.583). Conversely, OG-HFUS demonstrated a sensitivity of 91.8%, specificity of 96.0%, and almost perfect agreement (κ=0.858, CI: 0.763-0.952). Accordingly, their results demonstrated that OG-HFUS was a better tool than MSI in preoperatively estimating Breslow thickness and staging of cutaneous melanoma, emphasizing its potential as a valuable tool for melanoma diagnosis and treatment.

The study is interesting as it focuses on the possibility to non-inasively and preoperatively estimate the Breslow thickness of melanoma. The methods and the results are also well explained and discussed.

I have some concerns:

1. The authors must expand the discussion about the current non-invasive diagnostic tools for melanocytic and non-melanocytic skin diseases, including LC-OCT and reflectance confocal microscopy. The advantages and the limitations of this techniques in this context (melanoma and breslow thickness) must be better emphasize (for example the horizontal view of confocal is not useful fro estimating breslow thickness); please see PMID: 30357933 and PMID: 35666617 .

2. The fact that these non invasive techniques may be reliably used in clinical practise because of their strong correlation both with vertical and horizontal histopathology must be also discussed and emphasized. Please see the existing literature about this topic (correlation between psoriasis, discoid lupus and confocal and LC-OCT) 

Author Response

We extend our gratitude to the reviewer for their invaluable feedback. In response, we have enacted the following refinements:

1. We have expanded the Discussion section, providing a comprehensive overview of non-invasive diagnostic tools for melanocytic and non-melanocytic skin diseases. Specifically, we've broadened the scope of applications for LC-OCT and reflectance confocal microscopy (RCM). The studies referenced by the Reviewer have been deemed relevant and thoughtfully included in our discussion as follows “Moreover, Verzì et al.'s study illustrates the potential of LC-OCT to enhance the non-invasive…” (line 331)

2-We have emphasized and discussed the robust clinical applicability of these non-invasive techniques owing to their strong correlation with both vertical and horizontal histopathology. Additionally, we've referenced and emphasized existing literature regarding the correlation between psoriasis, discoid lupus, and the utilization of confocal and LC-OCT imaging modalities as follows “The discussion on the reliability of non-invasive techniques, such as LC-OCT and RCM, concerning their strong correlation with both vertical and horizontal histopathology in clinical practice is crucial. The existing literature, notably the study by Lacarrubba et al…” (line 367)

These adjustments notably improved the manuscript's thoroughness and coherence, and we extend our sincere gratitude for the valuable input given by the reviewer.

Round 2

Reviewer 2 Report

Comments and Suggestions for Authors

My concerns have been addressed.